# Data-driven analyses of behavioral strategies to eliminate cysticercosis in sub-Saharan Africa

Laura A. Skrip[1], Veronique Dermauw[2], Pierre Dorny[2], Rasmané Ganaba[3], Athanase Millogo[4], Zékiba Tarnagda[5], Hélène Carabin[6,7,8,9]*

1 University of Liberia, Monrovia, Liberia, 2 Department of Biomedical Sciences, Institute of Tropical Medicine, Antwerp, Belgium, 3 AFRICSanté, Bobo Dioulasso, Burkina Faso, 4 Department of Medicine, CHU Sourô Sanou, Bobo Dioulasso, Burkina Faso, 5 Institut de Recherche en Sciences de la Santé (IRSS), Bobo Dioulasso, Burkina Faso, 6 Département de pathologie et microbiologie, Faculté de médecine vétérinaire, Université de Montréal, St-Hyacinthe, Québec, Canada, 7 Département de médecine sociale et préventive, École de Santé Publique, Université de Montréal, Montréal, Québec, Canada, 8 Centre de recherche en Santé Publique (CReSP), Montréal, Québec, Canada, 9 Groupe de recherche en épidémiologie des zoonoses et santé publique, St-Hyacinthe, Québec, Canada

* helene.carabin@umontreal.ca

**Data Availability Statement:** All relevant data are within the manuscript and its Supporting Information files.

## Abstract

### Background

The multi-host taeniosis/cysticercosis disease system is associated with significant neurological morbidity, as well as economic burden, globally. We investigated whether lower cost behavioral interventions are sufficient for local elimination of human cysticercosis in Boulkiemdé, Sanguié, and Nayala provinces of Burkina Faso.

### Methodology/Principal findings

Province-specific data on human behaviors (*i.e.*, latrine use and pork consumption) and serological prevalence of human and pig disease were used to inform a deterministic, compartmental model of the taeniosis/cysticercosis disease system. Parameters estimated via Bayesian melding provided posterior distributions for comparing transmission rates associated with human ingestion of *Taenia solium* cysticerci due to undercooking and human exposure to *T. solium* eggs in the environment. Reductions in transmission via these pathways were modeled to determine required effectiveness of a market-focused cooking behavior intervention and a community-led sanitation and hygiene program, independently and in combination, for eliminating human cysticercosis as a public health problem (<1 case per 1000 population). Transmission of cysticerci due to consumption of undercooked pork was found to vary significantly across transmission settings. In Sanguié, the rate of transmission due to undercooking was 6% higher than that in Boulkiemdé (95% CI: 1.03, 1.09; p-value < 0.001) and 35% lower than that in Nayala (95% CI: 0.64, 0.66; p-value < 0.001). We found that 67% and 62% reductions in undercooking of pork consumed in markets were associated with elimination of cysticercosis in Nayala and Sanguié, respectively. Elimination of active cysticercosis in Boulkiemdé required a 73% reduction. Less aggressive reductions

**Funding:** This work was conducted with the support of the National Institute of Neurological Disorders and Stroke (NINDS) and of the Fogarty International Center (FIC) of the National Institutes of Health (NIH) under the Brain in the Developing World: Research across the life span program, grant R01NS064901 (to HC). The URL of the program is http://www.fic.nih.gov/Programs/Pages/brain-disorders.aspx. HC is funded in part by the Canada Research Chair in Epidemiology and One Health (CIHR/IRSC CRC 950-231857). The URL of the program is http://chairs-chaires.gc.ca. The funders had no role in study design, data collection and analysis, decision to publish, or preparation of the manuscript.

**Competing interests:** The authors have declared that no competing interests exist.

of 25% to 30% in human exposure to *Taenia solium* eggs through sanitation and hygiene programs were associated with elimination in the provinces.

## Conclusions/Significance

Despite heterogeneity in effectiveness due to local transmission dynamics and behaviors, education on the importance of proper cooking, in combination with community-led sanitation and hygiene efforts, has implications for reducing morbidity due to cysticercosis and neurocysticercosis.

## Author summary

It is important to consider context-specific behaviors and transmission pathways when designing scalable and sustainable intervention strategies for neglected tropical diseases (NTDs). To reduce the morbidity and mortality associated with cysticercosis, suites of interventions have been recommended but are inconsistently implemented due to cost and feasibility-related constraints. This study investigated the potential of a cooking intervention to interrupt transmission via undercooked pork in marketplaces of Burkina Faso. The sensitivity of *Taenia solium* parasite to temperatures attainable via improved cooking strategies provides a low-cost, human-centered approach to prevent consumption of infected pork meals. By accounting for differential behavior and the relative role of this transmission route across three provinces, we show how the potential of cysticercosis elimination (as a public health problem) varies across behavior-focused interventions. Further investigation into intervention strategies against human and pig cysticercosis warrants data-driven analyses that account for local variation in transmission behaviors.

## Introduction

Recognized by the World Health Organization as a neglected tropical disease, human cysticercosis has substantial health and economic consequences in populations dependent on subsistence farming for food and income [1]. Human cysticercosis is a condition that results as cysticerci, or larvae of the parasite *Taenia solium* (*T. solium*), develop in the body's tissues after people ingest *T. solium* eggs, and thus act as accidental intermediate hosts [2]. The usual life cycle of the parasite however involves pigs as intermediate hosts, and humans as final hosts. Pigs can develop cysticercosis upon ingestion of *T. solium* eggs, facilitated by free-roaming pig production practices and poor sanitation. Consumption of undercooked pork dishes prepared from infected pigs, in turn leads to *T. solium* taeniosis in humans, i.e. infection with the adult form of the tapeworm. Tapeworm carriers excrete parasite eggs when gravid proglottids are passed with the stool. Exposure to *T. solium* eggs, either in the environment, or through direct contact with taeniosis-infected individuals or self-infection, facilitated by open defecation, poor access to sanitation and hygiene, can lead to human cysticercosis. There is a general lack of data on taeniosis/cysticercosis generated from large epidemiological studies [3], although human cysticercosis has been reported in 22 countries across Africa and is considered endemic in most countries of Latin America and Southeast Asia [3,4]. In the high burden setting of Burkina Faso, the prevalence of active cysticercosis in villagers in 3 provinces was estimated at up to 11.5% [5].

*T. solium* has been estimated to incur 2.78 million DALYS in 2010, ranking number 4 out of 33 foodborne diseases, and having the higher rate of DALYs per 100,000 population in sub-saharan Africa of all foodborne infections evaluated [6]. When larvae of *T. solium* lodge in the central nervous system, a condition called neurocysticercosis develops and may cause symptoms including epilepsy, hydrocephaly, and stroke; the condition can be fatal [7,8]. Upon detection through brain imaging, neurocysticercosis may be treated with antiparasitic drugs or with anti-inflammatory drugs combined with symptom-specific treatment depending on the viability, number, and location of the cysts [9,10].

Intervention approaches for interrupting the taeniosis/cysticercosis transmission cycle have ranged from porcine vaccination [11] to ultrasonography diagnostic measures for pigs to improved sanitation infrastructure [12]. Less resource-intensive approaches have considered proper cooking strategies to remove the threat of infectious cysts from pork meat due to the sensitivity of adult *T. solium* parasites to temperature [13,14]. However, the lack of field evidence on which large-scale strategies are most sustainable and cost-effective has thwarted elimination of the morbidity associated with neurocysticercosis and financial losses associated with porcine cysticercosis [15].

Mathematical modeling has been increasingly used to inform intervention strategies against neglected tropical diseases in resource-constrained settings [16,17], although few models have been developed to simulate dynamic changes in the *T. solium* transmission cycle in response to human-, pig-, and/or environment-centered intervention programs [18]. Four methodologically unique transmission models [14,19–21] have been used to demonstrate the potential for attaining disease elimination across intervention scenarios. Three of these models [14,19,20] were implemented by incorporating epidemiological information from diverse settings or making hypothetical assumptions about generalized settings. Models informed by specific study settings could offer insight into context-sensitive intervention effectiveness.

We present a mathematical model of taeniosis/cysticercosis transmission to investigate the effect of a low-resource, human behavior-focused intervention, using field epidemiological data on disease prevalence and sanitation practices from three provinces of Burkina Faso. Specifically, we assess the impact of a cooking behavior intervention targeting market cookshop owners on reducing incidence of human cysticercosis. The cooking intervention was investigated alone and in combination with efforts to improve latrine use and basic hygiene. The model is the first to consider (1) the relative contributions to human cysticercosis of taeniosis carriers' autoinfection versus environmentally mediated mechanisms and (2) heterogeneity in intervention impact due to behavior-epidemiologic differences across geographically proximal, yet socio-demographically varying endemic settings. Our findings identify critical drivers of transmission and progression of cysticercosis, while highlighting gaps in current knowledge.

## Methods

### Ethics statement

This study was approved by the University of Oklahoma Health Sciences Center Institutional Review Board (IRB/1419) and the Centre Muraz Ethical Review Board (14-0027-AFRIC-SANTE/DR). It was also approved for the statistical analyses of the data by the Université de Montréal (#CERSES-19-081-D). Written consent was obtained from all participants or pig owners from whom data was used for this model. Formal consent was obtained by parent or guardian for children aged 18 years old or less, but children aged 10 or more were asked for their assent.

A deterministic compartmental model was developed to represent the multi-host taeniosis and cysticercosis infection system (Fig 1 and S1 Equations). The model was informed by

## Porcine host compartments

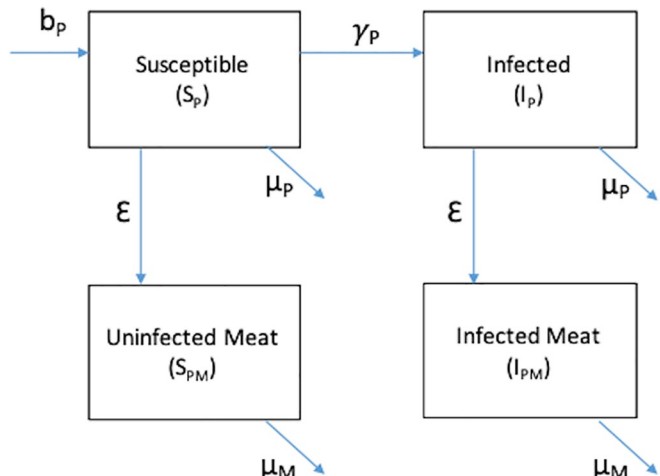

## Human host compartments

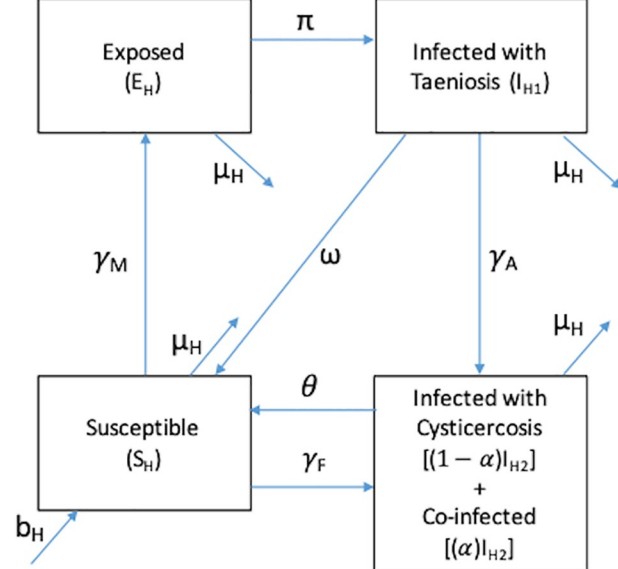

## Environment

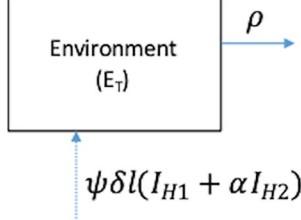

**Fig 1. Compartmental model structure for multi-host taeniosis and cysticercosis disease system.** A proportion of people with cysticercosis ($I_{H2}$) are assumed to be coinfected with taeniosis. Both autoinfection among those with taeniosis as well as consumption of infected pork by those with cysticercosis were recognized as mechanisms of coinfection.

epidemiological data collected as part of cohort and prevalence case-control studies conducted in Burkina Faso [22–24]. The multi-host transmission model was implemented using Matlab version R2016a (9.0.0).

## Data

**Description of the parent study.** Data collected as part of a large cluster-randomized controlled trial aimed at estimating the effectiveness of an educational intervention to reduce human and porcine cysticercosis were used to parameterize the dynamic transmission model. Specifically, data on human antibodies to *T. solium* adult stages measured during an embedded prevalence case-control study conducted at baseline [22] and on the prevalence of active cysticercosis in humans measured during the 18-month pre-randomization follow-up visit [22,24] were used.

The parent study was conducted between February 2011 and December 2014 in three provinces of Burkina Faso: Nayala, Boulkiemdé, and Sanguié. The baseline and pre-randomization visits took place from February 2011 to January 2012 and from August 2012 to July 2013 (*i.e.* 18 months apart), respectively. The selection criteria and procedures for study villages, households, concessions (*i.e.* a gathering of households in a compound) and participants for the baseline visit have been described in detail elsewhere [5,24]. Briefly, two villages present on official maps, at least 5 km apart from other study villages and with at least 1000 inhabitants, were randomly selected from each pig-raising department in the three provinces. In each of the 60 villages, a census was conducted to determine the number of concessions where sows and piglets were raised, the number of concessions without pig raising, as well as the number of households per concession. Eighty concessions were sampled according to a stratified random sampling approach, with type of pig raised at the time of sampling as stratum (*i.e.*, 10 concessions with sow raising, 30 concessions with piglet raising, 40 concessions with or without pig raising). In each selected concession, one household was randomly selected and one eligible individual (at least five years old, living in the village for at least one year, and not planning to move for the following three years) was randomly selected within each household and invited to participate in the study. In each village, sixty participants and all participants screening positive for epilepsy or severe chronic headaches at baseline were asked to provide a blood sample. During the pre-randomization follow-up, each participant was asked to provide a blood sample (60 per village) and to answer a socio-behavioral and screening questionnaire (20 additional subjects per village). In case the participant had moved away since the last visit, a person living in the same household was asked to participate in the study. At each visit, 40 pig owners (in the 40 randomly selected pig-raising concessions) were asked for their consent to take a blood sample on one of their pigs and were interviewed regarding their pig management methods.

## Sources of data for different elements of the dynamic transmission model

**Prevalence of active cysticercosis in humans.** The pre-randomization visit data were used to estimate the prevalence of active cysticercosis in the area. Mass drug administration of albendazole and ivermectin as part of a national lymphatic filariasis elimination program had been gradually phased out starting in 2012 [25], and the last community-wide delivery of

**Table 1. Data and terms informing the likelihood for the Bayesian melding model fitting, from pre-randomization visit, all groups.**

| Description | Equation using transmission model variables | Province-specific estimate | Sample size |
|---|---|---|---|
| **Humans** | | | |
| Prevalence of active cysticercosis in humans | $I_{H2}/N_H$ | B* = 6.7% (5.6–8.0); N* = 4.1% (2.7–5.9); S* = 3.4% (2.3–4.7) | B = 1555; N = 603; S = 844 |
| Prevalence of current taeniosis in humans (adjusted for ME) [a] | $[I_{H1}+\alpha\times(I_{H2})]/N_H$ | B = 3.5% (1.5–6.5); N = 2.6% (1.0–5.4); S = 2.3% (0.8–5.0) | |
| Prevalence of active cysticercosis/current taeniosis coinfection in humans (adjusted for ME) [a] | $[\alpha\times(I_{H2})]/N_H$ | B = 2.4% (1.0–4.3); N = 1.4% (0.5–2.8); S = 1.2% (0.4–2.3) | |
| **Pigs** | | | |
| Prevalence of active cysticercosis[b] | $I_P/N_P$ | B = 20.7% (4.2–47.8) N = 14.9% (1.0–40.5) S = 21.0% (4.3–48.8) | B = 857; N = 219; S = 643 |

*For B = Boulkiemdé; N = Nayala; S = Sanguié

[a] Estimates from Latent Class models, adjusting for misclassification error (ME)

[b] Uses calcified or degenerating cysts in specificity (Sp) estimation (considered as negative) and using priors from Chembensufo [26]

praziquantel for schistosomiasis control in the region occurred in 2010 (personal communication, health district data officers), while delivery to school-aged children every two years was ongoing in the region [24]. Our data showed an increase in the prevalence of active cysticercosis in the control group from baseline to the post-randomization visit [24], suggesting that the interruption of community-based delivery of MDA may have resulted in an increase in the prevalence of cysticercosis. Therefore, data from the pre-randomization visit, which were available for the 60 villages, were used as they may better reflect the endemic levels of cysticercosis in the population. We describe below the methods used to obtain the estimates needed to calibrate the model (Table 1).

Sera from 3,075 eligible and consenting participants providing a blood sample at the pre-randomization visit were used. Blood samples were obtained from the antebrachium vein and cooled until further processing. Sera were collected within three days after blood collection and frozen and stored at −20˚C until analysis. Serum samples were then tested for the presence of excretory–secretory circulating antigens of the metacestode of *T. solium* using the B158/B60 enzyme-linked immunosorbent assay (Ag-ELISA) [27]. This test has an estimated sensitivity of 90% (95% Bayesian credible interval [BCI]: 80; 99) and a specificity of 98% (95% BCI: 97; 99) for the detection of active human infection [28].

The prevalence of active cysticercosis was modeled as a binomial distribution with the number of individuals positive to the Ag-ELISA as the numerator and the total number of individuals tested in each province as the denominator.

**Frequency of food and sanitation practices.** Participants providing a blood sample also answered a questionnaire at the pre-randomization visit about sociodemographic factors and practices with regard to pork consumption, drinking water, sanitation, and self-reported tapeworm infection as well as knowledge of the life cycle of *T. solium* [22]. The distribution of sociodemographic factors and practices was similar between those who provided a blood sample and those who only answered the questionnaire. Therefore, data from the 3,002 participants who answered the questionnaire and provided a blood sample were used in the model.

**Pig demographics.** Rates of birth, natural death, and slaughter for pigs were derived from questionnaire data and personal communication with experts in swine medicine. From the

questionnaire data, it was determined that approximately 40% of concessions had pigs and that 20% of pigs were sows kept for reproduction. This estimate is consistent with a Food and Agricultural Report about the status of the pig production sector in Burkina Faso published in 2012 [29]. Out of 100 living pigs present in a village at any point in time, 27 will die due to other causes before slaughter, at 0.26 years of age on average, and 73 will die upon being sent to abattoirs for slaughter, at 1.64 years of age on average ([29], personal communication, Sylvie D'Allaire). Pig birth rate was determined to account for seven piglets per farrowing and 1.05 litters per sow per year, based on questionnaire data and expert knowledge.

**Prevalence of active cysticercosis in pigs.** At the pre-randomization visit, serum samples were obtained from 1,719 pigs. Blood samples were obtained from the jugular vein and the sera were frozen and stored at -20˚C until analysis with the Ag-ELISA described above. To estimate the prevalence of active cysticercosis in pigs, we had to take into account cross-reactions of *T. hydatigena* in the Ag-ELISA, due to its genus, not species-specific character [26]. We used data from a study where 452 pigs slaughtered in an abattoir located in Boulkiemdé province were inspected for the presence of *T. hydatigena*. This study found lesions of *T. hydatigena* in 8.8% of the inspected pigs [30]. This prevalence was similar to that found (10%) in a study conducted among 68 pigs of slaughtered age dissected as part of a study conducted in Zambia [31]. Therefore, we adjusted the Ag-ELISA results for the poor performance of the test in this setting using the sensitivity and specificity estimates found in the Zambian study. Using dissection as a gold standard, and considering the presence of only calcified cysts as negative for active cysticercosis, the Zambian study estimated a sensitivity of 91% (95%CI: 71% to 99%) and a specificity of 65.3% (95%CI: 48.5% to 77.3%) for the Ag-ELISA to detect active cysticercosis in pigs [31]. These values were used as priors in a Bayesian Latent Class model with one test and prevalence estimates adjusted for misclassification error were obtained using the method described in Joseph *et al.* 1995 [32]. The model was run in WinBugs 1.4.

**Prevalence of taeniosis in humans.** We used data from the baseline prevalence case-control study [21] to estimate the prevalence of taeniosis in humans. To this end, we used the proportions of participants seropositive to antibodies of taeniosis (*i.e.* exposed to taeniosis) among controls seropositive to active cysticercosis and those seronegative to active cysticercosis. This information was then used to estimate the proportion of the overall population currently with taeniosis alone or actively co-infected (*i.e.*, both taeniosis and active cysticercosis). The sampling of participants included in the prevalence case-control study has been described in detail elsewhere [22,33]. Briefly, a questionnaire was administered to all participants to screen for epileptic seizures, epilepsy and worsening severe chronic headaches. Sera from participants screened positive who agreed to provide a blood sample and from the same number of control participants matched by age group, gender and village were analysed for the presence of taeniosis antibodies using the rES33 test [34]. This test using recombinant antigens for detecting taeniosis antibodies has a reported sensitivity of 94.5% and specificity of 96% [34]. Because we were interested in current infection with taeniosis, we used a prior sensitivity value of 98% (95%CI: 96% to 100%) and a prior specificity value of 93% (95%CI: 86% to 100%) (John Noh, personal communication).

Among the 269 control participants, 14 were found positive to the rES33 test. This information was combined with the sensitivity and specificity prior information to obtain an adjusted estimate of the proportion of participants with active taeniosis among those Ag-ELISA positive and Ag-ELISA negative using a Bayesian Latent Class method described in Joseph *et al.* 1995 [32]. The adjusted proportions of current taeniosis among Ag-ELISA positive and Ag-ELISA negative controls were combined with data from the pre-randomization visit to obtain the prevalence of participants with current taeniosis only and current taeniosis and active cysticercosis. The model was run in WinBugs 1.4.

### Dynamic transmission model

**Model structure.**   A deterministic, compartmental model for transmission in the multi-host, cysticercosis-taeniosis system was developed as a set of differential equations, with separate compartment structures for pigs and humans (Fig 1 and S1 Equations). The host systems were linked through environmentally mediated transmission of *T. solium* eggs, which was explicitly represented with a compartment designated for simulating the availability of viable eggs to susceptible humans and pigs. Model parameters are presented in Table 2.

**Porcine hosts.**   Live pigs were represented in two states—susceptible ($S_P$) and infected ($I_P$)—and either died naturally at a rate $\mu_P$ or were slaughtered at a rate $\varepsilon$. Slaughtered pigs were uninfected ($S_{PM}$) or infected ($I_{PM}$) and transiently remained available as meat. The size of the live pig population was assumed to be constant, with the birth rate equal to the sum of the natural death rate and the slaughter rate. Derived from household-level survey data (See Data section in Methods), the initial conditions (Table 3) for the pig population were based on the mean number of pigs per household and the proportion of the human population reporting pig ownership.

Susceptible pigs become infected with cysticerci according to the force of infection ($\gamma_P$), which depends on the probability of transmission given ingestion of each *T. solium* egg ($\beta_E$) and the density-dependent rate of consumption of fecally contaminated soil ($\chi_P$):

$$\gamma_P = \beta_E \times \chi_P \times E_T, \tag{1}$$

where $E_T$ represents the number of viable eggs shed by humans with taeniosis and available in the environment, as described in a later section. Susceptible and infected pigs were assumed to be slaughtered at equal rates and their meat was likewise assumed to be consumed at the same rate.

No compartment was included to capture the state of pre-patency. Data from pig cysticercosis in the study area suggest that infection occurs early in the life of a pig in this setting and is ongoing. Indeed, it was observed that the prevalence was above 40% and did not increase after the age of 3 months. Therefore, even though there is a pre-patent period, most pigs that will become infected will have active cysts by age 1, 1.5 or 4 years old when they are slaughtered and consumed by humans.

**Human hosts.**   Initial human population conditions reflected available census data for the three provinces (Table 3). The human population was held constant, with equal birth and death rates, and was divided across compartments based on infection status for taeniosis ($I_{H1}$) or cysticercosis ($I_{H2}$). A proportion ($\alpha$) of those with cysticercosis were assumed to be co-infected with taeniosis. Co-infection was not explicitly modeled as a separate compartment due to lack of data on temporality of first versus second infections (Fig 1).

Pork-eating individuals susceptible to both taeniosis and cysticercosis ($S_H$) ingest *T. solium* cysticerci according to the frequency-dependent rate:

$$\gamma_M = \eta_M \times (I_{PM}/N_{PM}), \tag{2}$$

given the number of pigs with active cysticercosis slaughtered for meat ($I_{PM}$), the total number of pigs slaughtered for meat ($N_{PM}$) and $\eta_M$ is an expression for the rate of transmission due to undercooking of pork meat. We define the expression as

$$\eta_M = \beta_M \times [c_k \times (\sigma_{1k} + \sigma_{2k}) + c_h \times (\sigma_{1h} + \sigma_{2h})], \tag{3}$$

and where $\beta_M$ represents the probability of transmission of *T. solium* from pork meat to humans, where $\sigma$ and $c$ are the proportion of the population eating pork and the proportion of pork that is undercooked, respectively, in the market (subscript $k$) and at home (subscript $h$).

**Table 2. Parameter estimates for compartmental model of multihost cysticercosis-taeniosis system.**

| Parameter/ Quantity | Description | Raw Estimate[a] | Standardized monthly rate for parameter estimates | Detail, where applicable | Source(s) |
|---|---|---|---|---|---|
| $b_P$ | Birth rate for domestic pigs | 1.47 per pig per year | 0.123 per pig per month | 7 pigs per year per farrowing x 1.05 farrowing per sow per year x 20/100 sows per total pig population* | [42–43] |
| $\varepsilon$ | Slaughter rate for domestic pigs | 0.45 per pig per year | 0.038 per pig per month | See S1A Text and S1 Table for full explanation of derivation. 73% of pigs in the population will be slaughtered as follows: 60% are piglets with 10% sent to abattoir at age 1.5 years + 90% are piglets slaughtered at home at age 1 year + 40% are sows slaughtered at age 4 years | [26] |
| $\mu_P$ | Natural death rate for domestic pigs | 1.02 per pig per year | 0.085 per pig per month | Calibrated to achieve constant pig population | |
| $\mu_M$ | Average time of meat at market before sale | 2 days | 0.067 month | | |
| $b_H$ | Birth rate for humans (Burkina Faso) | 42.03/1,000 per year | 0.004 per month | | [44] |
| $\mu_H$ | Natural death of humans (Burkina Faso) | 42.03/1,000 per year | 0.004 per month | Set equal to the birth rate for closed model system | [44] |
| $\theta$ | Recovery after loss of cysticercosis infection | 48/109 per year | 0.037 per month | | [31] |
| $\pi$ | Development of mature tapeworm | 1/2.5 per month | 0.4 per month | | [1] |
| $\tau$ | Proglottids shed into the environment | 3 per day | 90 per month | | [2] |
| $\delta$ | Contribution of humans to environmental contamination | 55,000 x 3 per day | 4,950,000 per month | 55,000 fertile eggs per proglottid x τ proglottids per day per shedding human | [36] |
| $\omega$ | Natural resolution of taeniasis | 3/1000 per day | 0.09 per month | τ proglottids/1000 proglottids per worm per day per person | [2,35] |
| $\psi$ | Viability of eggs | 0.75 | | | [45–46] |
| $\alpha$ | Proportion of co-infected individuals (prevalence of co-infection / prevalence of active cysticercosis) | B: 35.8% N: 34.1% S: 35.3% | | Derived from Table 1 | [21] |
| $\sigma_{1k}$ | % of population USUALLY eating pork at a market in one's own village[b,c] | B: 8.1% (6.9–9.7) N: 2.7% (1.5–4.3) S: 6.1% (4.6–8.0) | | | [23] |
| $\sigma_{2k}$ | % of population USUALLY eating pork at another village's market[b,c] | B: 1.2% (0.8–1.9) N: 0.8% (0.3–1.9) S: 1.6% (0.8–2.7) | | | [23] |
| $\sigma_{1h}$ | % of population USUALLY eating pork at own home[b,c] | B: 48.5% (45.9–51.0) N: 48.5% (44.4–52.6) S: 59.0% (55.5–62.3) | | | [23] |
| $\sigma_{2h}$ | % of population USUALLY eating pork at another home in the same village[b,c] | B: 1.8% (1.2–2.6) N: 1.0% (0.4–2.2) S: 1.1% (0.5–2.0) | | | [23] |

(*Continued*)

**Table 2.** (Continued)

| Parameter/ Quantity | Description | Raw Estimate[a] | Standardized monthly rate for parameter estimates | Detail, where applicable | Source(s) |
|---|---|---|---|---|---|
| $l$ | % reporting ALWAYS using a latrine to defecate during the past 18 months[b] | B: 10.3% (7.8–11.9) N: 9.7% (7.4–12.4) S: 8.0% (6.2–10.0) | | | [23] |
| $\rho$ | Monthly rate of loss of viable eggs from environment | | | Fit | |
| $\chi_P$ | Monthly rate of consumption of fecally contaminated soil by pigs | | | Fit as the product $\beta_E\chi_P$ | |
| $\chi_F$ | Monthly rate among humans of ingestion of food or water contaminated by eggs in the environment and contact with eggs directly on another human carrier (collectively, $\chi_F$) | | | Fit as the ratio of $\chi_F/\chi_P$, assumed to be $<1$** | |
| $\chi_A$ | Monthly rate of cysticercosis infections among individuals with taeniosis due to autoinfection | | | Calibrated as the ratio of $\chi_A/\chi_P$, assumed to be $<\chi_F/\chi_P$ | |
| $\beta_E$ | Probability of transmission upon consumption of environmental contamination (*i.e.*, infected fecal matter) | | | Fit as the product $\beta_E\chi_P$ | |
| $\beta_M$ | Probability of transmission of *T. solium* from pork meat to humans | | | Fit as the product $\beta_Mc_k$ | |
| $c_k$ | Proportion of pork that is undercooked and consumed in the market | | | Fit as the product $\beta_Mc_k$ | |
| $\gamma_P$ | Force of infection for domestic pigs | *Eq 1* | | Province-specific | Calculated |
| $\gamma_M$ | Force of infection for humans consuming infected meat | *Eq 2* | | Province-specific | Calculated |
| $\eta_M$ | Rate of transmission due to undercooking of pork meat | *Eq 3* | | Province-specific | Calculated |
| $\gamma_F$ | Force of infection for humans consuming fecal contamination | *Eq 4* | | Province-specific | Calculated |
| $\gamma_A$ | Force of infection for humans consuming fecal contamination | *Eq 5* | | Province-specific | Calculated |

[a] All standardized to month$^{-1}$ during model implementation.

[b] Province-specific practice parameters using questionnaire data from the 3,002 participants providing a blood sample at the pre-randomization visit.

[c] For pork consumption estimates, data presented as % (95% Confidence Interval). Only the point estimate was included in the model for each parameter.

* Additional detail, including dimensional analysis for b$_P$, available in S1B Text.

** This assumption was based on the authors' understanding of behavioral exposure to soil contaminated with *T. solium* eggs for pigs versus humans. Since pigs spend the majority of time roaming to find their own food (in the dry season) or tethered (in the short wet season) with regular exposure to sources of environmental contamination, these hots were expected to have higher contact with *T. solium* eggs than humans, whose environmental exposure would be, for example, from latrine use, shaking contaminated hands with an infected individual, or consumption of contaminated food—behaviors being intermittent throughout the day and at dosage levels less than walking or laying on contaminated soil.

**Table 3. Province-specific initial conditions for population sizes.**

| Population | Province | | |
|---|---|---|---|
| | Boulkiemdé | Nayala | Sanguié |
| Human [a] | 700,924 | 227,112 | 410,555 |
| Pig [b] | 280,370 | 90,845 | 164,222 |

[a] Source: OCHA Regional Office for West and Central Africa.

[b] Assuming that 40% of concessions have pigs.

We assumed that pork consumed at home was always cooked at a temperature sufficient for killing the parasite. That is, we set $c_h = 0$. The assumption is consistent with information received at the study sites, as pork prepared at home is in the form of stews and boiled for long periods of time. Our data from interviews of mothers show that 98.5% of women preparing pork for the household boils the meat (3107/3156 mothers preparing pork). We further distinguished among the proportion of the population reporting most frequent pork consumption in markets in the same village (subscript $1k$) versus another village (subscript $2k$) and in their own homes (subscript $1h$) versus another home (subscript $2h$), with the former allowing for evaluation of village-level intervention that could be expected to only affect cooking behaviors within a village (described further in the section on *Intervention approaches*). Market-based pork meals in Burkina Faso include consumption of roasted meat (*porc-au-four*) or meat stews from street vendors and cookshops. Meat is often prepared in mud ovens for varying periods of time and at varying levels of heat.

Those individuals ingesting infected meat ($E_H$) will develop taeniosis and begin shedding *T. solium* eggs ($I_{H1}$) at a rate of $\pi$ which accounts for the period of parasite development. The rate of natural loss of infection will be dictated by the number of proglottids per worm and rate of shedding. Humans with taeniosis will naturally revert to being susceptible with a rate $\omega$ which corresponds to the days for all proglottids that an adult worm will produce in its lifetime to be shed.

Human may acquire cysticercosis via three routes (direct transmission through contact with taeniosis infected individuals, indirect transmission through the environment, and auto-infection), which differentially contribute to population-level prevalence. Infections through (1) direct contact with infected individuals (*e.g.*, through contact with eggs on the hands of a taeniosis carrier) or (2) indirect contact through the environment (*e.g.* contaminated food or drinking water) have been attributed to a higher number of infections than those associated with (3) direct self-infection, or autoinfection, among individuals with taeniosis.[1] Autoinfection is considered both in terms of real auto-infection (through reverse peristaltic movements in the bowel) and the self-infection due to poor hygiene (so-called external auto-infection).

The transition from the compartment of individuals susceptible to both taeniosis and cysticercosis ($S_H$) to that with individuals with cysticercosis ($I_{H2}$) is the result of environmentally mediated infections or direct contact with infected individuals. This density-dependent rate of cysticercosis transmission to entirely susceptible humans ($S_H$) depends on the probability of transmission given ingestion of each *T. solium* egg ($\beta_E$) and the combined rate among humans of ingestion of food or water contaminated by eggs in the environment and contact with eggs directly on another human carrier (collectively, $\chi_F$) and was given by

$$\gamma_F = \beta_E \times \chi_F \times E_T. \tag{4}$$

Cysticercosis among individuals with taeniosis ($I_{H1}$) is assumed to occur from autoinfection (at a rate $\chi_A$), direct transmission, or an environmentally mediated route (collectively, $\chi_F$) according to:

$$\gamma_A = \beta_E \times (\chi_A + \chi_F) \times E_T. \tag{5}$$

In addition, the pathway by which individuals with cysticercosis develop taeniosis was not modeled explicitly, although a small percentage of individuals were expected to experience co-infection through this route. The proportion ($\alpha$) of those with cysticercosis who also had taeniosis therefore reflected both individuals who transitioned from $I_{H1}$ to $I_{H2}$ at a rate $\gamma_A$ or who were already in $I_{H2}$ when exposed to contaminated pork meat. Both individuals with taeniosis

only ($I_{H1}$) and co-infected individuals contributed to the number of eggs shed into the environment as described in the next section.

Degeneration of all tissue cysts results in the movement of individuals to the susceptible compartment ($S_H$) at a rate $\theta$, which we define as loss of infection.

**Environmental compartment.** The $E_T$ compartment represents the number of viable eggs by which humans and pigs can become infected and develop cysticercosis. Individuals with taeniosis only or co-infected with taeniosis and cysticercosis are expected to release $\delta$ eggs, which depends on the number of proglottids shed and eggs released. Each person with taeniosis typically harbors one adult worm composed of 1,000 proglottids on average [35]. Up to five proglottids are shed each day with 30,000–90,000 eggs released per proglottid [36–38]. Only a percentage of released eggs are viable ($\psi$) and accessible for infecting hosts ($l$). Accessibility is related to the proportion of the population not reporting use of latrine or other improved sanitation facility for defecation (Table 2).

The number of viable and available eggs in $E_T$ is therefore given by

$$E_T = (I_{H1} + \alpha \times I_{H2}) \times \delta \times \psi \times l - E_T \times \rho, \tag{6}$$

where $\rho$ represents the monthly rate of decay of egg viability.

## Intervention approaches

The effectiveness of behavioral change intervention strategies in reducing the prevalence of active cysticercosis among humans was considered. Specifically, the interruption of *T. solium* transmission through pork cooking strategies and the interruption of exposure to *T. solium* eggs, both through environmentally mediated mechanisms and auto-infection, were modeled independently and in concert to assess whether behavior change could eliminate active cysticercosis from the study population.

Cooking to an adequate temperature was assumed to reduce the proportion of pork that is undercooked ($c$) by a factor of $E_M$. The reduction was specifically applied to the proportion of pork consumed in the markets of the village of residence ($\sigma_{1k}$) to account for a village-level intervention that may not extend to outside villages. That is the rate of transmission due to undercooking of pork meat (Eq 3), after accounting for the assumption $c_h = 0$, becomes $\beta_M \times c_k \times [(1-E_M) \times \sigma_{1k} + \sigma_{2k}]$.

Improvements in sanitation were assumed to be associated with behavioral changes in response to the education intervention used in the cluster randomized controlled trial of the parent study [23]. Improved sanitation was expected to reduce the rate of transmission associated with human exposure to *T. solium* eggs ($\beta_E$) by a factor of $E_E$. That is, the rate of transmission from eggs in the environment or through contact with someone with taeniosis to humans or to pigs becomes $(1-E_E) \times \beta_E$ with some hygiene or sanitation intervention.

For baseline (no intervention), the model was run to equilibrium using, as inputs, fixed parameters (Table 2) and acceptable posterior parameter sets identified during the fitting process and, as the output, the prevalence of active cysticercosis (per 1,000 population) at the equilibrium. For each intervention approach, the model was run to equilibrium with percent reductions in individual or multiple rates of transmission, according to our assumptions for how interventions would impact transmission, to represent the individual and combined intervention strategies, respectively, under consideration. The output of the intervention scenarios was likewise the prevalence of active human cysticercosis once the model reached equilibrium. In this way, we considered the long-term impact of routine and uniform implementation of interventions in endemic areas rather than the impact of pulse implementation or of time-dependent changes in implementation coverage.

For each intervention approach, the probability of reduction in prevalence to below a threshold of one case per 1,000 population was determined. In the absence of an elimination target from the World Health Organization, we defined this threshold for "elimination as a public health problem" [39]. The threshold was assessed using the average of the model output across the 1,000 sampled parameter combinations. Interventions were assumed to immediately and uniformly achieve a given reduction in transmission.

## Model fitting algorithm

The model was parameterized using published epidemiological data and data from Burkina Faso (See Data section of Methods, Table 2). For each of the three provinces ($j$), four quantities for which limited prior data exist and which were expected to vary by setting were fitted using a three-term log likelihood equation:

$$L(\beta_M c_k, \beta_E \chi_P, \chi_F / \chi_P, \rho) = \sum_{j=1}^{3} log[(n_j x_j) p_j^{x_j} (1 - p_j)^{(n_j - x_j)}] \tag{7}$$

based on the prevalence ($p$) of disease states derived from the model, as designated in Table 1. The three disease states were active cysticercosis in humans, current taeniosis in humans, and active cysticercosis in pigs (Table 1). A beta-binomial distribution was assumed, accounting for the number of individuals in the sample from the field data for disease state $j$ ($n_j$) and the number of individuals observed for the disease state $j$ ($x_j$). The four quantities sampled included the parameter for the monthly rate of decay of egg viability in the environment ($\rho$), as well as three aggregate quantities: (1) the product of the probability of transmission of $T$. *solium* cysticerci from pork meat to humans and the rate of undercooking in markets ($\beta_M c_k$), (2) the product of the probability of transmission given ingestion of each $T$. *solium* egg and the rate of consumption of fecally contaminated soil by pigs ($\beta_E \chi_P$), and (3) the ratio of the rate among humans of ingestion of food or water contaminated by eggs in the environment and contact with eggs directly on another human carrier versus the rate of pigs' consumption of fecally contaminated soil ($\chi_F / \chi_P$). The use of aggregate quantities was due to the lack of prior information to appropriately constrain the values of individual parameters for model identifiability.

A Bayesian melding approach [40] was used to randomly sample a set of parameter estimates from independent prior distributions and evaluate the model output according to the likelihood equation (Eq 7). Likelihoods were initially generated for 150,000 iterations of the model. The model was then implemented with 1,000 parameter sets, probabilistically sampled from the distribution of 150,000 according to the weight of their normalized negative log likelihood values. The melding approach was implemented separately for Boulkiemdé, Nayala, and Sanguié provinces. This procedure was repeated across a pre-defined set of parameter estimates for $\chi_A / \chi_P$: 0.0025, 0.005, 0.01, 0.025, 0.05, 0.10, 0.25, 0.5, and 0.75. For each province, $\chi_A / \chi_P$ was tuned between the two parameter estimates providing the best fit for further improvement. The mean of the log likelihoods generated from the posterior parameter sets for each $\chi_A / \chi_P$ value was sequentially compared with that for other $t$ values to determine which provided the best fit to the data.

For all iterations of model implementation, the model was run for 10,000 timesteps (*i.e.*, months) and the results were evaluated to assess whether endemic conditions had been reached. If the difference in prevalence of cysticercosis ($I_{H2}$) in the human population at t = 10,000 and t = 9,990 was <0.000001, the model was assumed to have reached dynamic equilibrium. If the condition was not met, the model was run for an additional 10,000 timesteps and re-evaluated. The process was repeated iteratively for a maximum 50,000 timesteps, and all iterations converged to equilibrium within that period.

We defined the rate of taeniosis transmission to humans upon consumption of infected pork by *Eq 3*. Therefore, the ratio of $\eta_{M(B)}/\eta_{M(S)}$ would represent the relative rate of transmission due to undercooking in Boulkiemdé as compared to that in Sanguié. The 95% Fiellers confidence interval [18] and a t-test for the ratio of means of two independent samples were calculated.

## Results

### Model fit and province-specific parameterization

On average, the model-predicted average prevalence of active human cysticercosis and current human taeniosis for all provinces and across parameter sets was reflective of the data, in terms of both point estimates and confidence intervals (Fig 2). Modeled estimates for cysticercosis in pigs generally included a wider range of values compared to those for human cysticercosis and taeniosis, although this observation was consistent with the 95% confidence intervals from the study data, which were also wider for the pig data than the human data (Table 1). Prior to intervention, the model-estimated average prevalence of human cysticercosis in Boulkiemdé, Nayala, and Sanguié was 6.7%, 4.5%, and 3.6%, respectively. Average prevalence of current human taeniosis in Boulkiemdé, Nayala, and Sanguié was estimated as 3.5%, 2.7%, and 2.3%. Estimates for average prevalence of active cysticercosis in pigs were 24.1%, 18.9%, and 23.7% in Boulkiemdé, Nayala, and Sanguié, respectively.

Province-level variations in the results of the model fitting process reflect the role of context-specific behavior in multi-host taeniosis and cysticercosis transmission dynamics (Fig 3). On average, the aggregate parameter ($\beta_E \chi_P$) was lowest in Boulkiemdé and comparable for Sanguié and Nayala (Distributions provided in Table 2). Given that $\beta_E$ is a biological parameter that could be assumed to be uniform across provinces, differences across the observed distributions for $\beta_E \chi_P$ were interpreted as relative, province-specific differences in rates of consumption of fecally contaminated soil by pigs ($\chi_P$). That is, pigs in Sanguié and Nayala were more often consuming *T. solium* eggs from the environment than were pigs in Boulkiemdé. Across all settings, the rate of human ingestion of food or water contaminated by eggs in the environment and contact with eggs directly on another human carrier was considerably lower than that of pig consumption of eggs in the environment—that is, $\chi_F/\chi_P << 1$. On average, the ratio $\chi_F/\chi_P$ was estimated to be lower in Sanguié than Boulkiemdé and Nayala. Given the previously stated observations about $\beta_E \chi_P$, human ingestion of food or water contaminated by eggs in the environment and contact with eggs directly on another human carrier was occurring at

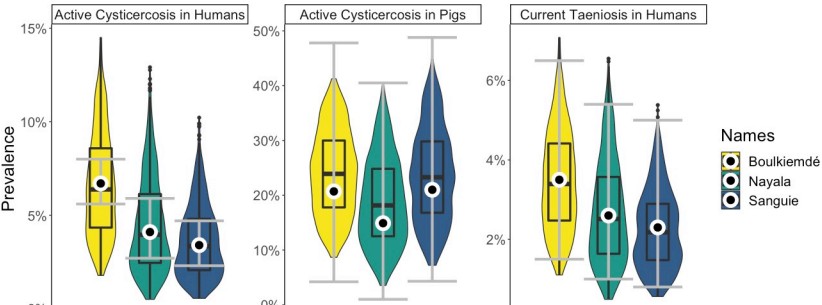

**Fig 2. Model fit to data.** The white circles and grey error bars are the point estimates and 95% CIs from the data. The violin plots (with embedded boxplots) are the model results. The point estimates, which were used for the likelihood, are close to the medians and means of the model output. The distribution of model output for cysticercosis in pigs and taeniosis in humans generally falls within the data CIs for those variables. The model is producing a larger range of values for active human cysticercosis than is observed in the data.

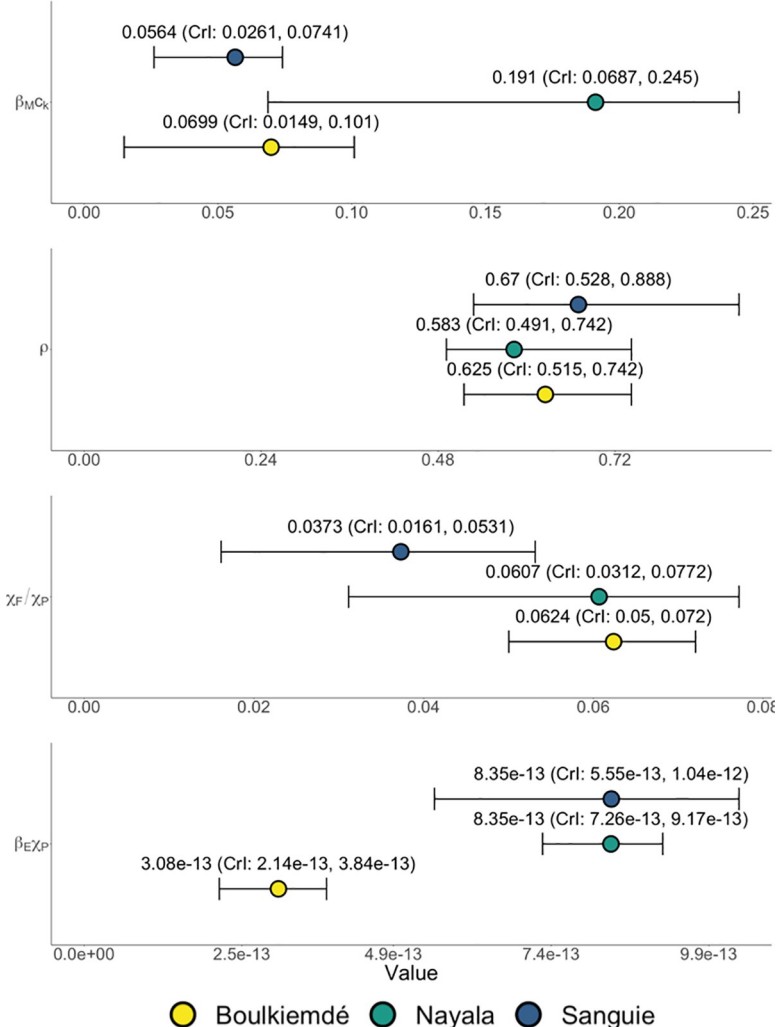

**Fig 3. Results of Bayesian melding model fitting procedure for the three study provinces in Burkina Faso: Sanguié, Boulkiemdé and Nayala.** Four, province-specific quantities (*i.e.*, monthly rate of decay of egg viability in the environment ($\rho$), the product of the probability of transmission of *T. solium* cysticerci from pork meat to humans and the rate of undercooking in markets ($\beta_M c_k$), the product of the probability of transmission given ingestion of each *T. solium* egg and the rate of consumption of fecally contaminated soil by pigs ($\beta_E \chi_P$), and the ratio of the rate among humans of ingestion of food or water contaminated by eggs in the environment and contact with eggs directly on another human carrier versus the rate of pigs' consumption of fecally contaminated soil ($\chi_F/\chi_P$)) were fit using data on three outcomes (*i.e.*, active cysticercosis in humans, active cysticercosis in pigs, and current taeniosis in humans).

a relatively lower rate in Sanguié than in Nayala. Compared to both Boulkiemdé and Nayala, $\chi_P$ in Sanguié was relatively greater than $\chi_F$. Moreover, the aggregate parameter ($\beta_M c_k$) was found to be higher in Nayala than in Boulkiemdé and Sanguié. As $\beta_M$ is a biological parameter, the observed variation is attributable to differences in the proportion of pork meals undercooked in markets ($c_k$), with the model suggesting that undercooking was more prevalent in Nayala than in the other provinces. The fitted parameter distributions for the rate of loss of viable eggs from the environment ($\rho$) were relatively consistent across provinces.

For Sanguié, the model best represented the data when the ratio of the rate of autoinfection to the rate of pigs' consumption of eggs in the environment ($\chi_A/\chi_P$) was 0.065. In contrast, for Boulkiemdé and Nayala, the model best represented the data when the ratio $\chi_A/\chi_P$ was 0.75

and 0.125, respectively. The calibrated values for $\chi_A/\chi_P$ correlated with the prevalence of current taeniosis—highest in Boulkiemdé and lowest in Sanguié.

### Transmission due to undercooking

The rate of taeniosis transmission to humans upon consumption of infected pork $\eta_M$ accounted for where pork consumption occurred, with undercooked meat assumed to only be available in market cookshops (*Eq 3*). Transmission due to undercooking in Nayala was estimated to be 1.63 times that in Boulkiemdé (95% CI: 1.59, 1.67; p-value < 0.001) and 1.54 times that in Sanguié (95% CI: 1.51, 1.57; p-value < 0.001). Transmission due to undercooking (*i.e.*, taeniosis infection due to consumption of infected pork) in Sanguié was found, on average, to be 6% higher than that in Boulkiemdé (95% CI: 1.03, 1.09; p-value < 0.001).

### Post-intervention model findings

In general, the prevalence of active cysticercosis in humans was more effectively reduced through interruption of exposure to *T. solium* eggs than through interruption of consumption of *T. solium* cysticercosis in undercooked pork meat for the same coverage levels (Fig 4).

In Sanguié and Nayala, elimination (*i.e.*, elimination as a public health problem) of active cysticercosis in humans was achieved, on average, upon a 62% and 67% reduction, respectively, in the rate of undercooking of pork meat. Elimination could be achieved with reduction of undercooking in Boulkiemdé by 73% (Fig 4A). More modest reductions of 25%, 26%, and 30% in human exposure to *T. solium* eggs, such as through community-led sanitation programs, were associated with cysticercosis elimination in Sanguié, Nayala, and Boulkiemdé, respectively (Fig 4B).

Intervention leading to at least 5% reduction in consumption of undercooked pork combined with less than 25% reduction in exposure to *T. solium* eggs in the environment was associated with elimination in Sanguié and Nayala (Fig 5A). Elimination in Boulkiemdé warranted at least 5% reduction in consumption of undercooked pork, combined with more than 25% but less than 50% reduction in environmental exposure to *T. solium* eggs. More aggressive campaigns against undercooking in markets, leading to 50% reductions in undercooking, led to elimination across all three provinces if combined with reductions of about 10% in environmental exposure to *T. solium* eggs. Boulkiemdé—where transmission due to undercooking was estimated as significantly lower than in Nayala and Sanguié—required more aggressive intervention to achieve elimination than the other provinces in all scenarios, while elimination was achieved in Sanguié with the least aggressive intervention.

## Discussion

The multi-host taeniosis/cysticercosis disease system includes multiple routes of transmission that afford opportunities for interrupting exposure to the different infectious stages of the parasite. Explicitly modeling the different hosts and routes of transmission allows for assessing targeted interventions in terms of their potential impact in eliminating human morbidity associated with cysticercosis. We found that reductions in transmission associated with interventions focused on changing behaviors, rather than introducing pharmaceutical tools or infrastructural innovations, is enough to eliminate active cysticercosis across different transmission settings in Burkina Faso. Combined approaches targeting both sanitation practices and cooking behaviors and henceforth interrupting multiple transmission routes achieved elimination at lower effectiveness levels than would be required by the individual interventions. Specifically, reductions of around 25% in exposure to *T. solium* eggs for both human and pig hosts, in combination with a 5% reduction in undercooking of pork in markets, were associated with universal elimination.

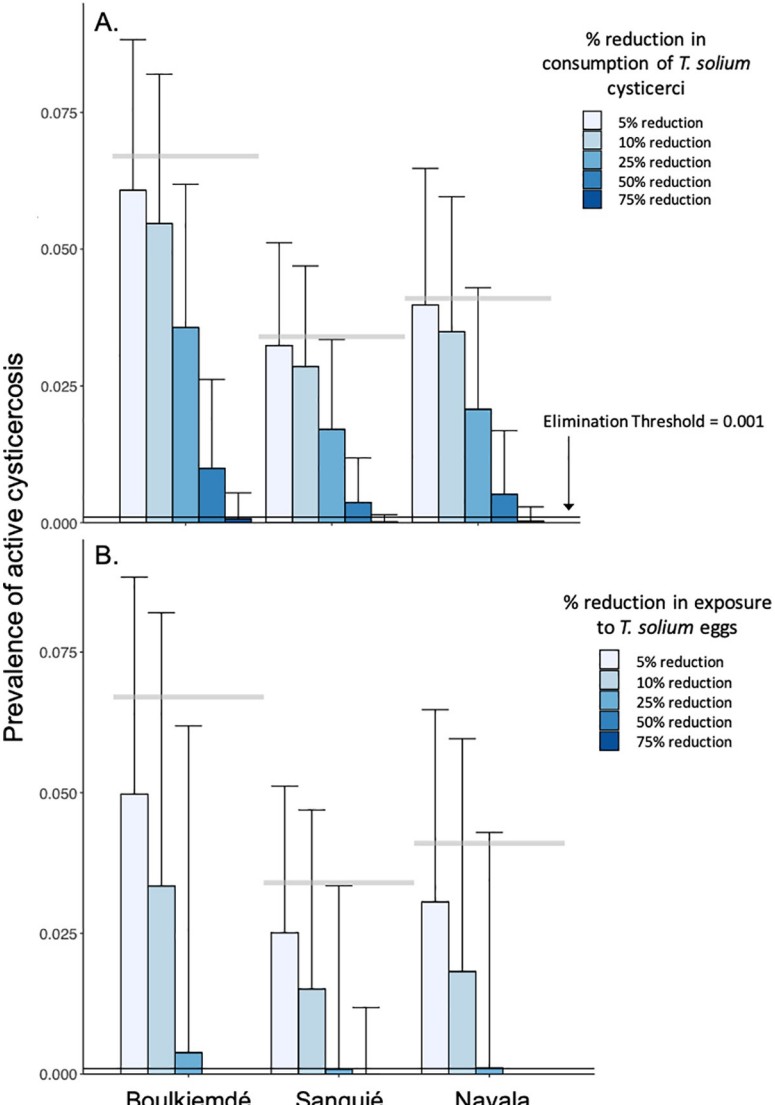

**Fig 4. Impact of behavioral change interventions on active cysticercosis infections. (A) Changes in the prevalence of active cysticercosis due to reductions in consumption of *T. solium* cysticerci in undercooked pork meat. (B) Changes in the prevalence of active cysticercosis due to reduced environmental exposure to *T. solium* eggs such as through use of latrines. Bars represent the average prevalence across simulations with 1,000 parameter sets.** Error bars represent one standard deviation above the mean. Horizontal, gray bars represent average, pre-intervention prevalence of active cysticercosis by province (See Table 1).

Existing modeling work on the public health burden of *T. solium* infection has increasingly been applied to consider a range of intervention strategies [17]. However, the lack of large-scale field data has led to assumptions about the effectiveness of the intervention and the base-line behaviors that are being intervened upon. Thus, existing models remain largely focused on theoretical populations. A recent review of mathematical models for *T. solium* identified gaps in modeling approaches and advocated for expanded, field-driven force-of-infection, population-based modeling [17]. Our model, including parameters used to inform province-specific practices around pork consumption and latrine use, extends existing work in a more field-informed and data-driven way to investigate potentially more logistically and

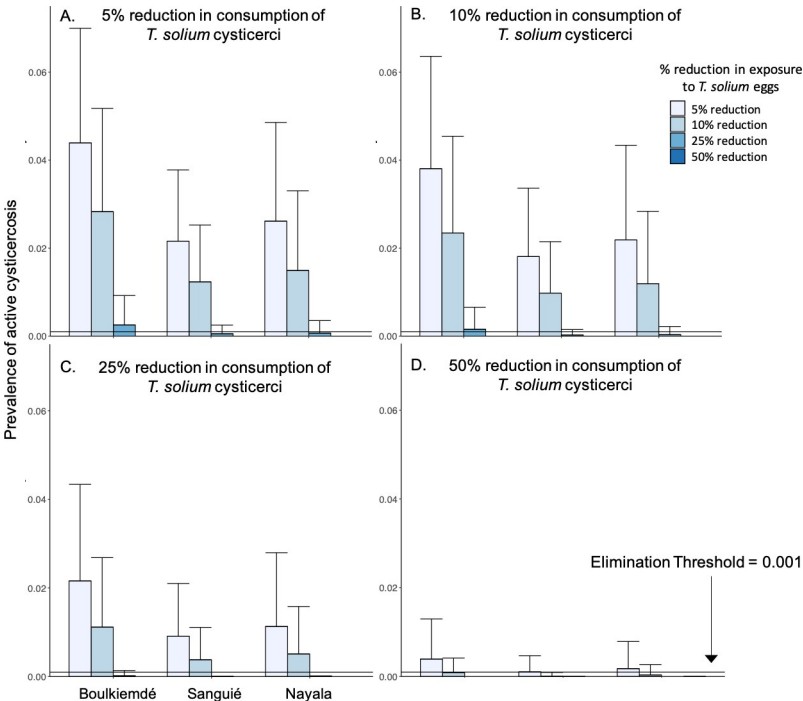

**Fig 5. Combined impact of pork cooking and latrine use interventions on active cysticercosis. (A) 5% reduction in exposure to *T. solium* cysticerci in undercooked pork meat, in combination with 5–50% reductions in exposure to *T. solium* eggs in the environment. (B) 10% reduction in exposure to *T. solium* cysticerci in undercooked pork meat, in combination with 5–50% reductions in exposure to *T. solium* eggs in the environment. (C) 25% reduction in exposure to *T. solium* cysticerci in undercooked pork meat, in combination with 5–50% reductions in exposure to *T. solium* eggs in the environment. (D) 50% reduction in exposure to *T. solium* cysticerci in undercooked pork meat, in combination with 5–50% reductions in exposure to *T. solium* eggs in the environment.** Bars represent the average prevalence across simulations with 1,000 parameter sets. Error bars represent one standard deviation above the mean.

economically feasible intervention. Our results emphasize the importance of accounting for local practices in assessing the potential of any intervention.

Boulkiemdé had the highest prevalence of active cysticercosis in humans and of taeniosis. Despite it being the highest burden setting, our findings suggest that transmission due to consumption of undercooked pork in market cookshops was less in Boulkiemdé than the other study settings per our posterior distributions for the aggregate parameter for the product of the probability of transmission of *T. solium* cysticerci from pork meat to humans and the rate of undercooking in markets ($\beta_M c_k$); since Nayala had a lower baseline prevalence of active cysticercosis in pigs than in Boulkiemdé, a higher rate of undercooking did not necessarily translate into a higher prevalence of taenioisis or human cysticercosis in Nayala. Likewise, the proportion of the population reporting latrine use to defecate was highest in Boulkiemdé. Thus, the investigated, human-focused interventions had incrementally less impact on disease transmission in this province than in Sanguié and Nayala. Pig-focused interventions could have greater impact in Boulkiemdé.

The results from this model become very useful in elucidating the observed effectiveness of the educational intervention tested as part of the clustered randomized controlled trial conducted in these villages. The trial showed that the intervention was most effective in Sanguié and Nayala, but not in Boulkiemdé [23]. The model suggested that more intensive improvements in latrine use and pork cooking habits were needed in Boulkiemdé than in Nayala and

Sanguié to eliminate the infection (Fig 4). Further investigation into the interplay between transmission dynamics and behavior across settings is warranted. The model has offered insights that can be incorporated into future field trials [41].

While we have aimed to establish a modeling framework that incorporates field data and expertise, it is recognized that the results presented here reflect simplifying assumptions that would benefit from additional epidemiological data, but also information on social aspects of the settings where the infection is present, in future iterations.

## Model limitations

The results presented here reflect several simplifying assumptions. For instance, future iterations of the model may account for changes in the pig and human populations over time. Also, the model did not account for spatial heterogeneity in the distribution of *T. solium* eggs throughout the environment and thus the differential exposure of humans and pigs to infection. There are currently no tests available to detect eggs in the environment (*i.e.* soil, water, objects, *etc.*), although incorporating it will remain a future direction for the model framework. Likewise, the current model does not account for heterogeneity in the burden of porcine cysticercosis infection. Without data on porcine parasite load or exposure to *T. solium* eggs (*i.e.*, rate of ingestion as the distribution of dosage across pig populations in each province), future work could introduce distributional assumptions, starting with consideration of an overdispersed distribution to reflect differential probability of exposure to high versus low dosages or differential burden of parasite load across the porcine population. Moreover, the force of infection in the pig population does not reflect age-specific contributions to transmission by infected pigs. Future iterations of the model, given available data, might be structured to capture age dynamics of transmission in pig hosts and not overestimate the contribution to transmission of slaughter-age pigs. While the added complexity may not be relevant in all cases, it could have implications for the evaluation of pig-targeted interventions. The sensitivity of the model results around human prevalence to distributional assumptions around differential burden or exposure among pigs, whether age-specific or population-level, could be explored.

The lack of data to inform individual parameters led to decisions around the structure and interpretation of our model. Specifically, the use of aggregate parameters was intended to maintain the model structure with all individual parameter components defined so that it can be used to answer more nuanced questions in the future, as additional data to inform components become available. However, by fitting aggregate quantities, interpretation of the parameter distributions—particularly the absolute values—is limited and it is difficult to pinpoint exact relationships across provinces for the individual component parameters.

Moreover, we modeled pulse intervention versus change in transmission through gradual interruption of exposure routes. The findings reflect expected reductions in prevalence of active cysticercosis when given percent reductions are attained. Our goal was to consider the potential utility of interrupting different transmission routes so that the feasibility and logistics of interventions with high potential for elimination of cysticercosis as a public health problem could be investigated by field experts to provide more information that could be fed into the model with future results iteratively reflecting logistical and epidemiological ground truths.

## Conclusions

Mathematical modeling of transmission within the multihost, taeniosis/cysticercosis system offers the potential to investigate novel intervention strategies working towards elimination of highly morbid neurocysticercosis. Efforts to address cooking and sanitation behaviors in settings with high prevalence of infected pigs and open defecation could successfully eliminate

the disease, as an alternative to treatment and vaccination approaches. Field data on the effectiveness of education campaigns in settings with prevalent cysticercosis would enhance future model findings to inform context-specific strategies.

## Supporting information

**S1 Equations. Transmission Model Equations.**
(DOCX)

**S1 Text. A** Description of Derivation of Slaughter Rate for Domestic Pigs ($\varepsilon$). **B** Dimensional Analysis for Birth Rate for Domestic Pigs ($b_P$).
(DOCX)

**S1 Table.** Description of Derivation of Slaughter Rate for Domestic Pigs ($\varepsilon$).
(DOCX)

## Acknowledgments

We thank Dr. Ida Sahlu for her help in generating village-level estimates and Assana Cissé-Koné and Dr. Sarah Gabriël for their contributions to analyzing the sera.

## Author Contributions

**Conceptualization:** Laura A. Skrip, Rasmané Ganaba, Athanase Millogo, Zékiba Tarnagda, Hélène Carabin.

**Data curation:** Veronique Dermauw, Rasmané Ganaba, Hélène Carabin.

**Formal analysis:** Laura A. Skrip.

**Funding acquisition:** Pierre Dorny, Rasmané Ganaba, Athanase Millogo, Zékiba Tarnagda, Hélène Carabin.

**Investigation:** Veronique Dermauw, Rasmané Ganaba, Athanase Millogo, Zékiba Tarnagda, Hélène Carabin.

**Methodology:** Laura A. Skrip, Veronique Dermauw, Rasmané Ganaba, Athanase Millogo, Hélène Carabin.

**Project administration:** Rasmané Ganaba, Athanase Millogo, Hélène Carabin.

**Resources:** Pierre Dorny, Rasmané Ganaba, Athanase Millogo, Zékiba Tarnagda.

**Supervision:** Rasmané Ganaba, Athanase Millogo, Hélène Carabin.

**Validation:** Laura A. Skrip, Veronique Dermauw, Hélène Carabin.

**Visualization:** Laura A. Skrip, Hélène Carabin.

**Writing – original draft:** Laura A. Skrip, Hélène Carabin.

**Writing – review & editing:** Laura A. Skrip, Veronique Dermauw, Pierre Dorny, Rasmané Ganaba, Athanase Millogo, Zékiba Tarnagda, Hélène Carabin.

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
