## [Decision Letter · Decision Letter 0]

5 Sep 2020

Dear Dr. Carabin,

Thank you very much for submitting your manuscript "Data-driven analyses of behavioral strategies to eliminate cysticercosis in sub-Saharan Africa" for consideration at PLOS Neglected Tropical Diseases. As with all papers reviewed by the journal, your manuscript was reviewed by members of the editorial board and by several independent reviewers. 

Our apologies for the time taken for our response, but the disruptions caused by the ongoing public health emergency r unanticipated delays in completion of the external reviews. In light of the reviews (below this email), we would like to invite the resubmission of a significantly-revised version that takes into account the reviewers' comments. 

We would like to see the issues mentioned by all the reviewers to be addressed, including the concerns raised by reviewer 3 regarding the model equations and the discussion.

We cannot make any decision about publication until we have seen the revised manuscript and your response to the reviewers' comments. Your revised manuscript is also likely to be sent to reviewers for further evaluation.

Sincerely,

Siddhartha Mahanty, M.B.B.S., M.P.H

Associate Editor

Marco Coral-Almeida

Deputy Editor

Reviewer's Responses to Questions

**Key Review Criteria Required for Acceptance?**

**Methods**

-Are the objectives of the study clearly articulated with a clear testable hypothesis stated?

-Is the study design appropriate to address the stated objectives?

-Is the population clearly described and appropriate for the hypothesis being tested?

-Is the sample size sufficient to ensure adequate power to address the hypothesis being tested?

-Were correct statistical analysis used to support conclusions?

-Are there concerns about ethical or regulatory requirements being met?

Reviewer #1: The manuscript "Data-driven analyses of behavioral strategies to eliminate cysticercosis in sub-Saharan Africa" offers an interesting and innovative methodology to study disease dynamic interactions based on community data. The reported results offer sufficient evidence to implement feasible strategies at a taeniasis/cysticercosis control program for researchers and decision-makers. However, I found that a small set of model parameters probably need to be corrected and probably redefined. Specifically in pig demography. Birth rate according to the data each sow gives birth 7 piglets per year, which means that for two years a sow will have 14 piglets and she will be 3 years old (so a probable rate 14/3 almost 5 times 0.2 because of adults, but this rate easily can be twice or even be triple value in commercial farms). Likewise, a dimensional analysis will be needed in the parameter definition.

Additonaly, the samplig strategy used would be better explained in a figure.

 It is necessary to mention that the model is a set of differential equations.

Running time should be mentioned to reach endemic conditions or at least mention the time when reported results were obtained.

Reviewer #2: -Are the objectives of the study clearly articulated with a clear testable hypothesis stated?

Yes.

-Is the study design appropriate to address the stated objectives?

Yes, given limitations and assumptions have been stated clearly (which they have).

-Is the population clearly described and appropriate for the hypothesis being tested?

Yes.

-Is the sample size sufficient to ensure adequate power to address the hypothesis being tested?

The data used in this study have been previously published. 

-Were correct statistical analysis used to support conclusions?

Yes, to the best of my knowledge.

-Are there concerns about ethical or regulatory requirements being met?

No.

Reviewer #3: The methods are generally clearly outlined and suitable for the research questions outlined. I do have some important questions/points that require clarification (see comments).

**Results**

-Does the analysis presented match the analysis plan?

-Are the results clearly and completely presented?

-Are the figures (Tables, Images) of sufficient quality for clarity?

Reviewer #1: They are clear and completely presented, but they have to be confirmed after parameter corrections.

Reviewer #2: -Does the analysis presented match the analysis plan?

Yes

-Are the results clearly and completely presented?

Yes – with a few amendments to be made.

-Are the figures (Tables, Images) of sufficient quality for clarity?

Yes – with a few amendments.

Further comments:

• Line 356 should refer to average prevalence of active human cysticercosis to align with Figure 2(a) label.

• Figure 2 is a nice illustration/comparison of modelled outputs compared to data. Consider changing the y-axis to percentages for readability (e.g. 0%, 5%, 10%, 15% instead of 0, 0.1, 0.15). 

• Results are clearly presented including any clearly outlining assumptions made regarding interpretation of results.

• Figure 3 needs a legend to indicate which province is represented by blue, yellow and green estimates. 

• Figure 5 caption does NOT align with the figure. E.g. looking at Figure 5A and the labels therein I would interpret this as: 5% reduction in exposure to T. solium cysticerci in combination with a 5 – 50% reduction in exposure to T. solium eggs. This is NOT what is written in the Figure 5 caption on lines 443 – 452. The same comment goes for Figures 5B, 5C and 5D. 

From the discussion text I think that it is the figure 5 caption which is incorrect.

• How was the elimination threshold of less than one case per 1,000 population decided upon?

Reviewer #3: The analysis matches the objectives outlined in the introduction. See further comments for clarifications in the results section.

**Conclusions**

-Are the conclusions supported by the data presented?

-Are the limitations of analysis clearly described?

-Do the authors discuss how these data can be helpful to advance our understanding of the topic under study?

-Is public health relevance addressed?

Reviewer #1: Is it possible to mention which pork-made dishes are prepared and why they could contain undercooked pork?

Reviewer #2: -Are the conclusions supported by the data presented?

Yes

-Are the limitations of analysis clearly described?

Yes

-Do the authors discuss how these data can be helpful to advance our understanding of the topic under study?

Yes

-Is public health relevance addressed?

Yes

Reviewer #3: The conclusions are generally well supported. This paper provides an important contribution in the area of T. solium research, and the authors explain how this advances our understanding in the field.

**Editorial and Data Presentation Modifications?**

Reviewer #1: Minor modifications:

L44 and elsewhere, citations and references are after final sentence dot.

L201: mention that the model is a set of differential equations

L335 is Eq 7?

Reviewer #2: (No Response)

Reviewer #3: Table 1: It is not clear what ME is, could the authors please define this.

Table 1: What does Sp represent in the footing of Table 1 (line 133) – I think it refers to susceptible pigs as as per the model figure, but this has not been defined yet in the text.

Table 2: It would be easier for the reader if another column was made available in Table 2 to indicate the standardized monthly rate for parameter estimates. 

In Figure 3, it would be useful to include a legend for colour of the points based on each province for quick reference (otherwise there is a need to constantly refer back to Fig 2 to check)

Line 335: this should refer to equation 7 for the likelihood equation

**Summary and General Comments**

Reviewer #1: The manuscript "Data-driven analyses of behavioral strategies to eliminate cysticercosis in sub-Saharan Africa" offers an interesting and innovative methodology to study disease dynamic interactions based on community data. The reported results offer sufficient evidence to implement feasible strategies at a taeniasis/cysticercosis control program for researchers and decision-makers. The data offered by authors probabbly is unique to estimate worthing information of epidemiological parameters.

Reviewer #2: This article presents a deterministic multi-host (humans and pigs) compartmental model for the transmission of different stages of the tapeworm Taenia solium in a closed population. The purpose of the study presented is to consider the impact of less resource intensive interventions focused on human behaviours to eliminate transmission as a public health problem. The authors point out the importance of utilising large field-based study data to inform such interventions. 

Overall, I think this study is of public health relevance and is a good starting point to addressing the gaps in currently available models of taeniosis/cysticercosis and the impacts of interventions on elimination of the disease as a public health problem. I have detailed comments below which need to be addressed prior to the publication of this manuscript.

The authors refer to elimination and define it as less than one case per 1,000 population. My understanding would be that this refers to “elimination as a public health problem”, if this is the case the authors should clarify this within their article. 

The article is lacking a clear and concise description of what the authors refer to as “taeniosis/cysticercosis” (note that in the author summary only ‘cysticercosis’ is referred to). This needs to be incorporated more clearly into the introduction to give readers a clear idea of the disease in question . E.g. incorporate a description of the parasite lifecycle and the link between taenoiosis and cysticercosis – this could be written description or inclusion of a figure such as the one here: https://www.cdc.gov/parasites/cysticercosis/biology.html

Some of this information is presented under the dynamic transmission model section, but it needs to be explained earlier in the article than this to ensure you don’t lose readers, for example by referring to parasite stages (eg proglottids) before explaining why those stages are important.

• General comment: at first mention of Taenia solium put in brackets (T. solium) before using abbreviation throughout. Do not assume the reader will know what the abbreviation stands for.

• Question from assumption on line 227: do the authors think it is it realistic to assume that susceptible and infected pigs were slaughtered at equal rates, or are infected pigs likely to show signs of illness and be culled earlier or removed from production? 

• Thorough description of the transmission model. The authors provide a transparent description of assumptions made about model parameters and the way they have incorporated the impact of interventions on reducing transmission. 

• Line 332: Comment on the limitation or impact the use of aggregate quantities may have on derived estimates and model outcomes? 

• Line 335: typo, I assume the likelihood equation is actually Equation (7) on line 320?

Reviewer #3: My summary: This is an important paper that, as identified by the authors, contributes to filling an gap in the area of T. solium research (developing transmission models informed by field-data). The model also provides a novel contribution by robustly assessing the impact of non-pharmaceutical interventions, with most T. solium models to date focusing on pharmaceutical interventions. In terms of contribution to understanding prospects for control, the paper considers interventions which might be more feasible/sustainable in severely resourced endemic settings. 

The approach adopted uses data from a detailed study in Burkina Faso (with pre-randomisation baseline prevalence data, adjusted for diagnostic performance in a latent class model). A Bayesian Melding model fitting approach samples key parameters including the rate of loss of viable eggs (which is an important parameter to elucidate as very minimal data is available), two transmission coefficients for human taeniosis and porcine cysticercosis, and the ratio of human contact rate with eggs with pig contact rate with eggs. After fitting, the contact rates are identified as higher in Nayala compared to the other two provinces, while the rate of loss of viable eggs is similar across settings. Reduction in exposure to T. solium eggs is presented as a more effective intervention, compared to reducing consumption of undercooked meat at reaching human active cysticercosis elimination thresholds. 

The differences in modelled contact rates between provinces eloquently explain the different effectiveness of modelled interventions between provinces, supporting the observations of heterogeneous intervention effectiveness in the field trial conducted in Burkina Faso by the same group. 

Overall, I very much like this manuscript and think it will be an important contribution. I do however has some important points to raise below. 

Introduction:

Including data/text on burden of disease in the introduction paragraph 2 (lines 46-50) to help contextualize the clinical aspect for more general audiences. 

Methods:

Table 1

If I am understanding the methodology correctly, the prevalence estimates provided in Table 1 as the adjusted prevalence estimates obtained from the Latent class model, which are then used for model fitting. If this is the case, highlighting in Table 1 that these are adjusted prevalence estimates (not unadjusted), would improve clarity. 

Table 2

It is not clear to me how the slaughter rate for domestic pigs is calculated from the detail provided (and how the reference [24] Chembensofu et al. 2017 is used to inform this). Please could the authors clarify. 

Equations and structure

Could the authors explain why a pre-patent compartment in pigs is omitted, where this duration (2- 3 months as demonstrated by Yoshino, 1933a,b)) is non-negligble compared to the relatively short life-span of pigs in endemic settings :

Yoshino K. Studies on the post-embryonal development of Taenia solium, Pt. i. J Med Assoc Formosa 1933a;32:139–41. 

Yoshino K. Studies on the Postembryonal Development of Taenia solium. Part II. On the Youngest Form of Cysticercus cellulosae and on the Migratory Course of the Oncosphaera of Taenia solium within the Intermediate Host. J Med Assoc Formosa 1933b;32:1569-1586.

Could the authors explain their rationale behind not including heterogeneity in burden of porcine cysticercosis infection (e.g. low or high burden) which could influence the probability of infection for human taeniosis infection?

The force-of-infection equation for human taeniosis (equation 2) indicates that human contact rate with pork is assumed to be frequency-dependent. It would be useful to explicitly state this, and similarly for human or pig contact rates with contaminated soil, where equations 1 and 4 indicate density-dependent contact rates are assumed. 

Could the authors clarify the assumption that pork consumed at home “ was always cooked at a temperature sufficient for killing parasites” on lines 244-245 compared to pork cooked in the market. Did they see this in their data/ study area?

I am not clear on the definition of seroreversion rate parameter (θ) on lines 280-281, and Table 2 - is this seroreversion from both human cysticercosis and human taeniasis seropositivity back to full susceptibility? Lines 280-281 makes me think this is only seroreversion from human cysticercosis? Please could the authors clarify. 

Also, should θ be defined as the human recovery or infection loss rate (from true infection i.e IH2 to SH), as presumably the seroprevalence data is adjusted based on the diagnostic performance, so seroreversion can be interpreted as those who are truly reverting from infected status to susceptibility? I find the name of the parameter (the seroreversion rate) slightly confusing as the model captures true infection dynamics (not seropositivity)?

How are the interventions applied? Are they implemented as an instantaneous reduction in the rate of transmission by the factors mentioned on lines 301 and 308 and maintained throughout the duration of the intervention programme? Also, when is the modelled prevalence measured to assess the impact of the intervention – is this in line with the 18-month follow-up (post randomisation survey) indicated in ref [22]? This is not clear in the “Intervention approaches” section of the methods. There is a little text on modelling “pulse interventions” in the model limitations section (lines 502-503), however more information should be provided in the methods section. 

SI Model equations

There appears to be an error in equation # 4 in the SI equation: should the addition term to dIpm/dt be εIP not εIPm?

It is not clear what the subscript j refers to in model fitting algorithm section (lines 317 – 325), presumably where the model was fitted to each province, j represents each province? If so, could this be explicitly written (i.e. “For each of the provinces (j), four..”

Could the authors provide justification for the assumption that the contact rate of pigs with eggs in the environment (χP) is always larger than the contact rate of humans with eggs in the environment/ with other taeniosis carriers (χF) or autoinfection (χA), given the parameter estimates outlined in line 340 for model fitting and the information outlined in Table 2?

Results:

I find the Figure 5 legend slightly confusing – is 5A not a 5% reduction in consumption rather than 20% in the legend text on line 444 for example?

Discussion: 

This section is currently quite short, so there is space for expansion on some points. Specific questions include:

The authors mention the added value of this paper in presenting “field-driven force-of-infection” modelling. It would be useful if the authors could present in the manuscript the specific province-level FoI estimates calculated in the model at endemic equilibrium. 

Do the authors have any idea why the modelled contact rates are higher in Nayala, while the pre-intervention prevalence is the highest in Boulkiemdé? Would it not expected that a higher baseline T. solium prevalence would be found in a setting with higher contact rates (Nayala)? The authors begin to elude to this in lines 477-480, but I think more discussion should be provided. 

The rate of loss of viable eggs in the environment is an important parameter, where minimal data is available from field settings (most models relying on data from other Taenia species). In Figure 3, the authors find a similar rate of loss of viable eggs in environment in each province. Is this expected? perhaps the authors could discuss more about the similarity (or not) of environmental conditions across provinces which could lead to these findings. 

In results, it is stated that "βE is a biological parameter that could be assumed to uniform across provinces", however the probability of viable infection upon contact (with infective material) may vary on the basis of whether pigs are exposed to smaller doses e.g. contacting eggs in the soil (lower probability) compared to higher doses e.g. consuming proglottids (higher probability). Santamaria et al. 2002 (ref below) indicate that the efficiency of establishment (proportion of viable cyst developing) is influenced by the dose (eggs consumed). Have the authors considered this? 

Santamaria E, Plancarte A, de Aluja AS. The Experimental Infection of Pigs with Different Numbers of Taenia solium Eggs: Immune Response and Efficiency of Establishment. J Parasitol 2002;88:69.

PLOS authors have the option to publish the peer review history of their article (what does this mean?). If published, this will include your full peer review and any attached files.

Reviewer #1: Yes: Lenin Ron Garrido

Reviewer #2: No

Reviewer #3: No
---

## [Decision Letter · Decision Letter 1]

13 Jan 2021

Dear Dr. Carabin,

Thank you very much for submitting your manuscript "Data-driven analyses of behavioral strategies to eliminate cysticercosis in sub-Saharan Africa" for consideration at PLOS Neglected Tropical Diseases. As with all papers reviewed by the journal, your manuscript was reviewed by members of the editorial board and by several independent reviewers. The reviewers appreciated the attention to an important topic. Based on the reviews, we are likely to accept this manuscript for publication, providing that you modify the manuscript according to the review recommendations. 

The reviewers acknowledged and were satisfied by the content and detail of your responses. We would like to see a few remaining and reasonable questions stemming from your responses addressed. We encourage you to address was many of the questions as you can, but particularly the query regarding point 28 requested by Reviewer #3, if possible. 

Sincerely,

Siddhartha Mahanty, M.B.B.S., M.P.H

Associate Editor

Marco Coral-Almeida

Deputy Editor

Reviewer's Responses to Questions

**Key Review Criteria Required for Acceptance?**

**Methods**

-Are the objectives of the study clearly articulated with a clear testable hypothesis stated?

-Is the study design appropriate to address the stated objectives?

-Is the population clearly described and appropriate for the hypothesis being tested?

-Is the sample size sufficient to ensure adequate power to address the hypothesis being tested?

-Were correct statistical analysis used to support conclusions?

-Are there concerns about ethical or regulatory requirements being met?

Reviewer #1: The new version of the article has incorporated important recommendations suggested for the previous version. I think methodological observations were taken into account and they were addressed correctly. Maybe the software used to run the analysis should be mentioned in the Methodology. Consequently, I consider that the stated questions were properly answered and therefore, the article must be accepted.

Reviewer #2: (No Response)

Reviewer #3: (No Response)

**Results**

-Does the analysis presented match the analysis plan?

-Are the results clearly and completely presented?

-Are the figures (Tables, Images) of sufficient quality for clarity?

Reviewer #1: Results offer important insights for the human Taeniasis-cysticercosis control at the community level. The model captures the general Taeniasis-cysticercosis community dynamics and its parameter estimates seem adequate in the transmission dynamics.

Reviewer #2: (No Response)

Reviewer #3: (No Response)

**Conclusions**

-Are the conclusions supported by the data presented?

-Are the limitations of analysis clearly described?

-Do the authors discuss how these data can be helpful to advance our understanding of the topic under study?

-Is public health relevance addressed?

Reviewer #1: The conclusion presented by the authors in the manuscript reflects clearly what they were looking for in the objectives of this research. It also gives important insights for control strategies against the Taeniasis -cysticercosis disease complex, therefore it is of public health relevance.

Reviewer #2: (No Response)

Reviewer #3: (No Response)

**Editorial and Data Presentation Modifications?**

Reviewer #1: Authors properly have managed the questions related to the estimation of epidemiological parameters and therefore, I think, the model may be a useful tool with proper parameter values necessary to understand this disease dynamics and to manage strategies for disease control and elimination in the communities. Therefore I recommend accepting it to publish in PLOS NTD. I have suggested just to include the software used for the model running.

Reviewer #2: (No Response)

Reviewer #3: (No Response)

**Summary and General Comments**

Reviewer #1: The manuscript "Data-driven analyses of behavioral strategies to eliminate cysticercosis in sub-Saharan Africa" provides important epidemiological parameters useful to model the epidemy dynamics and to predict the disease behavior with control tools for this disease complex. The model suggests that the use of suitable tools at the community level may help control the problem and gives insights about the period and necessary efforts to reduce the burden of the disease. The authors have properly have discussed the drawbacks of this kind of modeling, and have described the main model limitations. In spite of this, model conclusions result epidemiologically relevant.

Reviewer #2: By revising this manuscript and addressing concerns highlighted by the reviewers in detail, the authors have produced an interesting manuscript of public health relevance which is now fit for publication.

Reviewer #3: Firstly, I would like to thank the authors on their detailed responses, and congratulate the authors on an improved manuscript. Before proceeding, I would like to first request clarification on the following responses please:

In response to point 27:

Thank you for further clarification on the slaughter rate, this is much clearer now. Could the authors walk me through their derivation of 12;14;74% dying for the three different reasons mentioned in the supplementary paragraph below:

"Among all pigs alive at any point in time, 74% are pigs who will survive to slaughter. This is based on the following: among 7 piglets born per sowyear (based on local expertise (Ganaba), we assume that in the rural setting each sow of reproductive age farrows once per year and the number of piglets is based on ref [29]), 50% will die before weaning at age 0.17 years (based on ref [29] and our data), 20% will die at age 0.5 years between weaning and slaughter (death rate in this period is not available but assumed from the age distribution of pigs alive at the time of our surveys), and 30% will survive to slaughter (including sows kept for reproduction) corresponding to 12%, 14% and 74% of piglets dying before weaning, dying between weaning and

slaughter and being slaughtered (including sows kept for reproduction)."

In response to point 28:

If I am understanding correctly, does this not indicate that non-infected pigs largely become resistant after age of 3 months (with continued exposure), if (age-?)prevalence does not increase from 40% after 3-months of age (assuming constant exposure with age)? As this is not an age-structured model or does not include parameters for immunity, the authors will not be able to plot an age-prevalence curve to see if the model captures these age-infection dynamics. However, the model presented without age-structuring here presumably models a constant force-of-infection with respect to age, therefore the prevalence will increase with age. I want to authors to comment on this, given they mention in the manuscript the prevalence doesn’t rise from 3 months of age and their model will not presumably capture their own field observations (of prevalence saturating with age; therefore the current model assumes prevalence keeps increasing with age, so slaughter age-animals will have a higher prevalence than those at 3-months of age), and have not included parameters for immunity (which may lead to an underestimation in the force-of-infection). See Lewis and Torgerson 2014 (Echinoccocus multilocularis), and Torgerson et al 1998 (Taenia hydatigena) which use a simpler catalytic model formulation fitted to age-prevalence curves for further reference. 

- Lewis, F.I., Otero-Abad, B., Hegglin, D., Deplazes, P. and Torgerson, P.R., 2014. Dynamics of the force of infection: insights from Echinococcus multilocularis infection in foxes. PLoS neglected tropical diseases, 8(3) e2731. 

- Torgerson, P.R., Williams, D.H. and Abo-Shehada, M.N., 1998. Modelling the prevalence of Echinococcus and Taenia species in small ruminants of different ages in northern Jordan. Veterinary Parasitology, 79(1), pp.35-51.

In response to points 32 & 33:

I still find this this a little unclear- as the authors response suggest, can they authors remove the “from seropositivity” and “(seroreversion) in the “Recovery from seropositivity after loss of cysticercosis infection (seroreversion)” for parameter θ in table 2, and similarly on line 313 of the revised manuscript, removing “seroreversion”. 

In response to point 34:

With the intervention simulations run to equilibrium, can the authors state the timeframe(s) to reach equilibrium under the different intervention strategies (is there significant variation for example if I am understanding correctly)? Are these within a realistic timeframe (i.e. if the simulations have taken a significant time to reach equilibrium, is it realistic that interventions/behaviour changes would be maintained for so long)?

Additional comment:

Will model code be available upon publication, which will be important for transparency (for example, see other published Taenia solium transmission models, CystiSim and EPICYST which both have code available through GitHub).

PLOS authors have the option to publish the peer review history of their article (what does this mean?). If published, this will include your full peer review and any attached files.

Reviewer #1: No

Reviewer #2: No

Reviewer #3: No
---

## [Editor Report · Decision Letter 2]

10 Feb 2021

Dear Dr. Carabin,

We are pleased to inform you that your manuscript 'Data-driven analyses of behavioral strategies to eliminate cysticercosis in sub-Saharan Africa' has been provisionally accepted for publication in PLOS Neglected Tropical Diseases.

Best regards,

Siddhartha Mahanty, M.B.B.S., M.P.H

Associate Editor

Marco Coral-Almeida

Deputy Editor

---

## [Editor Report · Acceptance letter]

17 Mar 2021

Dear Dr. Carabin,

We are delighted to inform you that your manuscript, "Data-driven analyses of behavioral strategies to eliminate cysticercosis in sub-Saharan Africa," has been formally accepted for publication in PLOS Neglected Tropical Diseases.

Best regards,

Shaden Kamhawi

co-Editor-in-Chief

Paul Brindley

co-Editor-in-Chief
